# Diuretic treatment before and after transcatheter aortic valve implantation: A Danish nationwide study

**Xenia Begun**[1]*, **Jawad Haider Butt**[1], **Søren Lund Kristensen**[1], **Peter Ejvin Weeke**[1], **Ole De Backer**[1], **Morten Schou**[2], **Lars Køber**[1], **Emil Loldrup Fosbøl**[1]

1 Department of Cardiology, Rigshospitalet, Copenhagen University Hospital, Copenhagen, Denmark,
2 Department of Cardiology, Herlev-Gentofte University Hospital, Hellerup, Denmark

* Xeniabegun@gmail.com

**Data Availability Statement:** All relevant data are within the paper and its Supporting Information files.

## Abstract

### Objectives

We examined loop diuretic treatment before and 1-year after transcatheter aortic valve implantation (TAVI), as a proxy for changes in symptom severity and secondly assessed how changes in loop diuretics related to mortality risk.

### Background

Randomized clinical trials suggest that approximately one third of patients undergoing TAVI do not achieve symptom relief, but "all-comer" data are lacking.

### Methods

Using Danish nationwide registries, we identified all citizens, who underwent TAVI from 2008 to 2019 and were alive at 1-year post-discharge. Loop diuretic treatment pre-TAVI and at 1-year post-TAVI were assessed and grouped as receiving 1) no-loop diuretics; 2) low: 1–40 mg of furosemide (or equivalent bumetanide) daily; 3) intermediate: 41–120 mg of furosemide daily; or 4) high: >120 mg furosemide daily.

### Results

Among the 4431 patients undergoing TAVI, 2173 (49%) patients were not treated with loop diuretics at the time of TAVI, 918 (21%) had low-loop diuretics, 881 (20%) had intermediate-loop diuretics, and 459 (10%) had high-loop diuretics. At 1-year post-TAVI, 893 (20%) patients had increased, 1010 (23%) had reduced, and 2528 (57%) had unchanged loop diuretic treatment. The cumulative 5-year risk of death in patients surviving one year, was 61% (95% CI: 56.4% to 65.3%) in patients with increased and 47% (95% CI: 44.9% to 49.9%) in patients with reduced/unchanged loop diuretic treatment, respectively. In multivariable Cox proportional hazard analysis, increased loop diuretic treatment was associated with a higher risk of death compared with reduced/unchanged loop diuretic treatment (Hazard ratio: 1.4; 95% CI: 1.22 to 1.52).

**Funding:** The author(s) received no specific funding for this work.

**Competing interests:** The authors have declared that no competing interests exist.

## Conclusions

Among patients undergoing TAVI, surviving one year, one fifth of patients had increased loop diuretic treatment, and a little over one fifth had reduced loop diuretic treatment 1-year post-procedure. In patients with increased diuretic treatment, the risk of death was higher compared to those with reduced/unchanged loop diuretic treatment.

## Introduction

Randomized clinical trials in aortic valve replacement (AVR) suggest that about one third of patients with severe aortic stenosis (AS) do not have symptom relief after undergoing transcatheter aortic valve implantation (TAVI) [1, 2]. Current guidelines suggest AVR as a definitive treatment of AS, and so far, no medical therapies influence the natural history of AS [3, 4]. In recent years, TAVI has revolutionized treatment of AS, and it has surpassed surgical aortic valve replacement (SAVR) and has become the standard of care in patients of older age with AS [5]. Still, only few studies have evaluated the quality of life of patients undergoing TAVI, including symptom relief [6–8]. Studies have shown TAVI to be a life-extending and non-inferior treatment option in high, intermediate and low surgical risk groups compared with SAVR [9–12]. However, despite the rapidly increasing number of procedures, data on life quality and symptom relief after TAVI procedure from "all-comer" cohorts are sparse [13]. Patients with severe symptomatic AS present with a range of clinical symptoms, all with high impact on quality of life, and to grade symptom burden, the standard procedure is to use New York Heart Association (NYHA) functional class, which classifies the degree of dyspnea. Especially dyspnea, which appears due to vascular congestion, affects the quality of life of the patients, and loop diuretic treatment is often used to relieve symptoms and improve quality of life [14]. Randomized trials have shown a significant reduction in symptoms, lower NYHA class and fewer cardiac symptoms after undergoing TAVI [2]. However, knowledge about identification of patients who do not benefit from TAVI in terms of symptom relief is limited. This emphasizes the need for further studies to identify and characterize these patients.

The objective of this study was to evaluate loop diuretic treatment prior and 1-year post-TAVI, as a proxy for symptom severity, and examine baseline factors associated with increased and reduced loop diuretic treatment in order to investigate symptom burden and quality of life after intervention with TAVI and relate changes in diuretic treatment to all-cause mortality.

## Methods

### Data sources

A unique and permanent personal civil registration number is assigned to all Danish citizens, which enables linkage at an individual level of nationwide registries. Information on medication was obtained through The Danish Registry of Medicinal Product Statistics (National Prescription Registry) which holds information on all dispensed drug prescriptions from 1995, all classified according to the Anatomical Therapeutical Chemical (ATC). Detailed information on all claimed drug prescriptions in Denmark including date of dispensing, quantity dispensed, strength and the affiliation of the physician issuing the prescription, is comprised in the registry. The registry yields a high validity due to pharmacies in Denmark are required to register all prescriptions dispensed to get coverage by the healthcare system financed by the

government [15]. The Danish Civil Registration System includes information on sex, birth date, vital status, whether a person is emigrated or dead, including dates of these events [16]. To get information on comorbidity and hospital admissions, data from Danish National Patient Registry was obtained, which holds information on all hospital admissions to Danish hospitals since 1978 according to the International Classification of Diseases (until 1994 the 8th revision (ICD-8) and after 1994 the 10th revision (ICD-10)) and Nordic Medico-Statistical Committee Classification of Surgical Procedures. Each admission is registered by one primary diagnosis and if appropriate, one or more secondary diagnoses [17].

## Study population

The study population comprised all Danish citizens undergoing first-time TAVI between January 1st 2008 and December 31st 2019 surviving one year following discharge from hospital after TAVI procedure. The index date for the present study was defined as the date of hospital discharge after TAVI procedure. Furthermore, a nationwide cohort of patients with heart failure and loop diuretic treatment undergoing TAVI, was examined. All patients, who died within 12 months after the index date, were excluded from the study.

## Dose and treatment duration

Information on diuretic treatment was obtained by identifying all claimed prescriptions from the Danish National Prescription Registry for loop diuretics: furosemide (ATC-code C03EB01) or bumetanide (ATC-code C03EB02) and for non loop diuretics: thiazides (ATC-code C03A, C03B, C07B, C07D and C09XA52, C03EA01) and spironolactone (ATC-code C03DA01, C03DA02, C03DA03, C03DA04). For loop diuretic treatment, prescriptions were identified from 6 months prior to the date of admission for TAVI (pre-TAVI loop diuretic treatment) and 1-year post-TAVI-index date (post-TAVI loop diuretic treatment) were identified. The National Prescription Registry does not comprise information on daily dose and use of loop diuretics can change on daily basis. Prescriptions were used to calculate daily dose. Five prescriptions prior to the actual prescription were assessed, constituting a treatment interval, and hereafter estimating an average dose according to the prescription information on quantity dispensed and strength of the prescription. Dosages were allowed to change, and to estimate the duration of the treatment, the dispensed number of pills was divided by the calculated daily dose, which is described in details elsewhere [18, 19]. The calculated daily dose of loop diuretic treatment was used to define the loop diuretic treatment pre-TAVI and loop diuretic treatment post-TAVI, and patients were divided into following groups according to the daily dose of the claimed drug: 0) no loop diuretic treatment; 1) low: 1–40 mg of furosemide or < = 1 mg of bumetanide; 2) intermediate: 41–120 mg of furosemide or 2–3 mg bumetanide and 3) high: >120 mg furosemide or >3 mg bumetanide. Non-loop diuretic treatment, including thiazides and spironolactone, were defined by having at least one claimed prescription 6 months prior to the date of admission of TAVI and in the period 6 months to one year after TAVI index date.

## Comorbidity and concomitant pharmacotherapy

Comorbidity was obtained by identifying hospitalizations any time before discharge date after the TAVI procedure (Admissions were identified by using ICD-8 and ICD-10 diagnosis codes) except individuals with hypertension and diabetes, which were identified using claimed drug prescriptions [20, 21].

## Outcomes

The primary outcomes were change in loop diuretic treatment pre-TAVI and at 1-year post-TAVI and all-cause mortality among patients alive one year after discharge for TAVI procedure. Increased, reduced, and unchanged loop diuretic treatment were defined as a change from one pre-TAVI loop diuretic group to a higher, lower, or identical post-TAVI loop diuretic group, respectively. Furthermore, secondary outcome was a combined outcome consisting of change in loop diuretic treatment and number of hospital readmissions for heart failure within 1-year post-procedure. The 5-year cumulative risk of death in patients surviving one year was compared according to the two groups of patients, who 1) had increased loop diuretic use or 2) had reduced/unchanged loop diuretic treatment. Patients, who survived one year after discharge for first-time TAVI procedure, were followed from 1-year post-procedure until death, emigration, or end of study (31 December 2019), whichever came first.

## Statistics

Baseline characteristics were summarized as frequencies with percentages, means or medians with 25th-75th percentiles. Differences in baseline characteristics between loop diuretic dosage groups were examined using Cochran-Armitage trend test for binary variable and Cochran-Mantel-Haenszel test for categorical variables. The cumulative incidence of death from 1-year post-TAVI was calculated using the Aalen-Johansen estimator according to increased or reduced/unchanged loop diuretic treatment 1-year post-TAVI. Differences between the two groups were assessed using Gray's test. The Kaplan-Meier method was used to construct survival curves and differences between groups were assessed using log-rank test. Hazard ratios (HR) with 95% confidence intervals (CIs) were estimated using multivariable Cox proportional hazard regression. The model for all-cause mortality was adjusted for age (included as a categorical variable: <76 years, 76–80 years, 81–85 years and 86 < = years), year of procedure (included as a categorical variable: 2008–2010, 2011–2013, 2014–2016 and 2017–2019), sex, a history of heart failure, ischemic heart disease, stroke, atrial fibrillation, hypertension, peripheral vascular disease, diabetes, malignancy, chronic kidney disease, chronic obstructive pulmonary disease, and liver disease. Logistic regression analysis was used to identify covariates associated with increased diuretic treatment, and odds ratios (ORs) with 95% confidence intervals were calculated for available covariates associated with increased loop diuretic treatment. The proportional hazards assumption was tested and found valid. All statistical analyses were performed with SAS 9.4 statistical software (SAS Institute, Cary, North Carolina, USA) and R. For all analyses, a two-sided p value <0.05 was considered statistically significant.

## Ethics

The Danish Data Protection Agency approved the study [approval number: P-2019-191]. In Denmark, retrospective registry-based studies, that are conducted for the sole purpose of statistics and scientific research, do not require ethnical approval.

## Results

In total, 4849 citizens underwent first-time TAVI between 1 January 2008 and 31 December 2019. Of these, 418 (8.6%) patients died during admission or within 12 months post-discharge, and were excluded from the study, leaving 4431 patients (Fig 1). At time of TAVI procedure, 1277 (29%) patients were not on diuretic treatment, 1474 (33%) had only loop diuretic treatment, 896 (20%) had non loop diuretic treatment, and 784 (18%) had both loop and non loop diuretic treatment. The median age of the study population was 81.5 (25th-75th percentile 77–

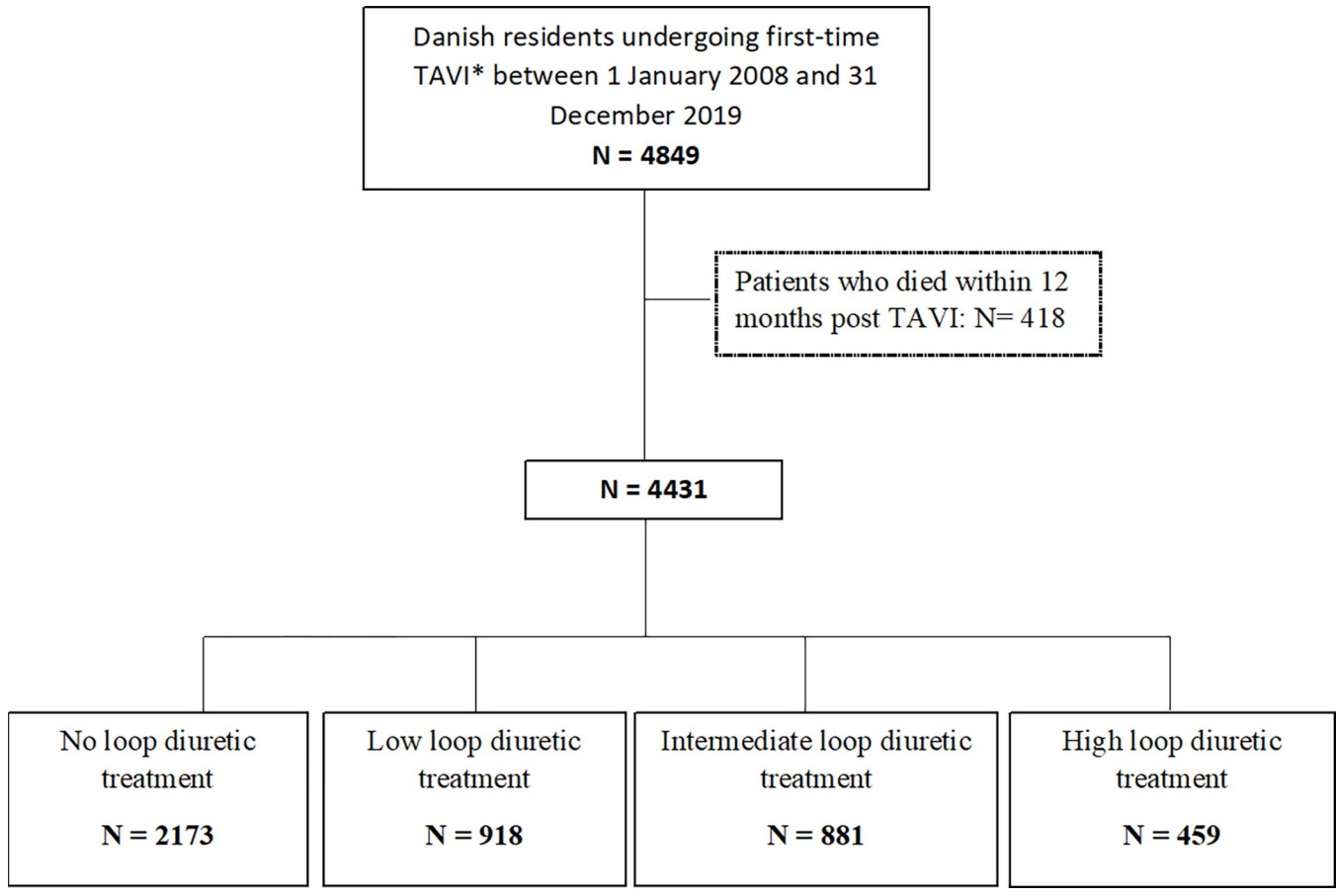

**Fig 1. Flow chart of the study population selection process.** Abbreviations: TAVI, transcatheter aortic valve implantation. Low loop diuretic treatment 1–40 mg of furosemide (or equivalent bumetanide); intermediate loop diuretic treatment 41–120 mg of furosemide; high loop diuretic treatment >120 mg furosemide.

85) and 54% were men. Loop diuretic treatment pre-TAVI and 1-year post-TAVI according to daily loop diuretic dosage are illustrated in Fig 2. Baseline characteristics divided into groups according pre-TAVI loop diuretic treatment are shown in Table 1. Patients with pre-TAVI high and intermediate loop diuretic treatment had a greater burden of cardiovascular and non-cardiovascular comorbidities compared with patients with low or no loop diuretic treatment. The distribution of age and sex was similar between groups. In total, 893 (20%) patients had increased, 1010 (23%) patients had reduced, and 2528 (57%) patients had unchanged loop diuretic treatment at 1-year post-TAVI. Outcomes including number of HF-admissions within 1-year post-procedure are depicted in Table 2. The cumulative 5-year risk of death in patients undergoing TAVI surviving one year was 61.0% (95% CI: 56.4% to 65.3%) in patients with increased loop diuretic treatment and 47.4% (95% CI: 44.9% to 49.9%) in patients with reduced/unchanged loop diuretic treatment 1-year post-TAVI (Fig 3A). Additionally, the cumulative 5-year risk of death in patients surviving one year that were on diuretic treatment at TAVI procedure, was 64.7% (95% CI: 58.4% to 70.3%) in patients with increased loop diuretic treatment and 56.2% (95% CI: 52.8% to 59.4%) in patients with reduced/unchanged loop diuretic treatment (Fig 3B). The cumulative 5-year risk was also examined according to the three groups separately (S1 Fig in S1 File). In multivariable Cox proportional hazard analysis, increased loop diuretic treatment was associated with a higher risk of death (HR = 1.36,

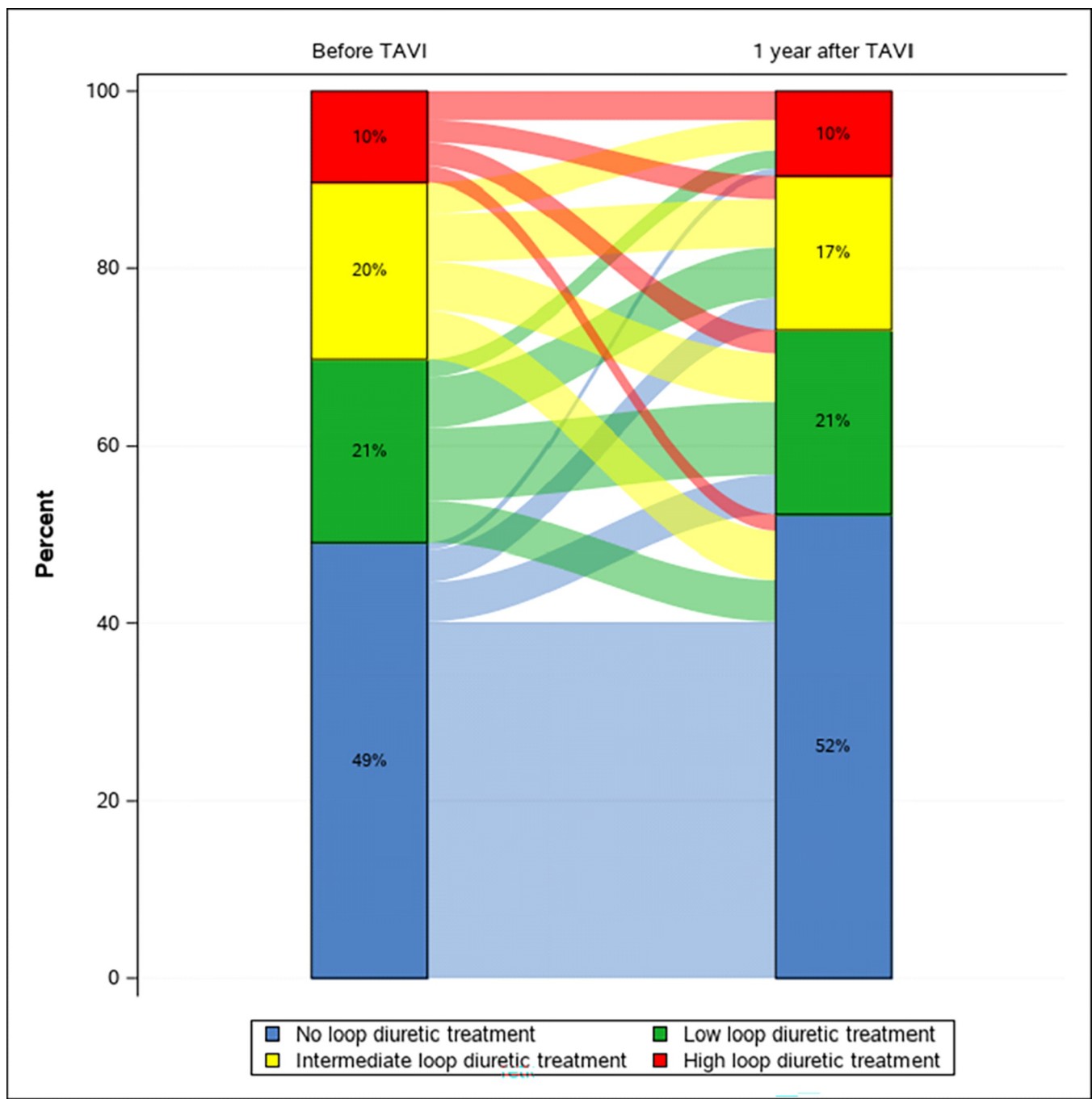

**Fig 2. Changes in loop diuretic groups pre-TAVI and 1-year post-TAVI.** Abbreviations: TAVI, transcatheter aortic valve implantation. Low loop diuretic treatment 1–40 mg of furosemide (or equivalent bumetanide); intermediate loop diuretic treatment 41–120 mg of furosemide; high loop diuretic treatment >120 mg furosemide.

95% CI 1.22–1.52) compared with reduced/unchanged loop diuretic treatment. In multivariable Cox proportional hazard analysis for patients, who were on diuretic treatment pre-TAVI, increased loop diuretic treatment was associated with a higher risk of death (HR = 1.31, 95% CI 1.13–1.52) compared with reduced/unchanged loop diuretic treatment.

**Table 1. Baseline characteristics of the study population.**

| | No loop diuretic treatment n = 2173 | Low loop diuretic treatment n = 918 | Intermediate loop diuretic treatment n = 881 | High loop diuretic treatment n = 459 | P-value |
|---|---|---|---|---|---|
| **Demographics** | | | | | |
| Age, yrs | 80.0 (77–85) | 81.1 (78–85) | 81.5 (78–86) | 80.5 (76–86) | <0.0001 |
| Male | 1156 (53.2%) | 515 (56.1%) | 472 (53.6%) | 251 (54.7%) | 0.56 |
| **Comorbidities** | | | | | |
| Heart failure | 320 (14.7%) | 393 (42.8%) | 398 (45.2%) | 283 (61.7%) | <0.0001 |
| Ischemic heart disease | 1081 (49.7%) | 503 (54.8%) | 497 (56.4%) | 271 (59.0%) | <0.0001 |
| Stroke | 313 (14.4%) | 203 (22.1%) | 188 (21.3%) | 122 (26.6%) | 0.38 |
| Atrial fibrillation | 569 (26.2%) | 397 (43.2%) | 402 (45.6%) | 246 (53.6%) | <0.0001 |
| Peripheral vascular disease | 148 (6.8%) | 69 (7.5%) | 79 (9.0%) | 50 (10.9%) | 0.001 |
| Hypertension | 299 (13.8%) | 134 (14.6%) | 135 (15.3%) | 66 (14.4%) | 0.21 |
| Diabetes | 325 (15.0%) | 176 (19.2%) | 189 (21.5%) | 128 (27.9%) | <0.0001 |
| Malignancy | 448 (20.6%) | 162 (17.6%) | 204 (23.2%) | 97 (21.1%) | 0.34 |
| Chronic kidney disease | 132 (6.1%) | 93 (10.1%) | 104 (11.8%) | 109 (23.7%) | <0.0001 |
| Chronic obstructive pulmonary disease | 280 (12.9%) | 154 (16.8%) | 174 (19.8%) | 91 (19.8%) | <0.0001 |
| Liver disease | 54 (2.5%) | 18 (2.0%) | 32 (3.6%) | 12 (2.6%) | 0.29 |
| **Concomitant pharmacotherapy** | | | | | |
| Thiazides | 844 (38.8%) | 246 (26.8%) | 194 (22.0%) | 84 (18.3%) | <0.0001 |
| Spironolactone | 78 (3.6%) | 117 (12.8%) | 121 (13.7%) | 72 (15.7%) | <0.0001 |
| Aspirin | 1064 (49.0%) | 452 (49.2%) | 407 (46.2%) | 234 (51.0%) | 0.84 |
| ADP-receptor inhibitors | 530 (24.4%) | 227 (24.7%) | 214 (24.3%) | 119 (25.9%) | 0.64 |
| Beta-blockers | 927 (42.7%) | 503 (54.8%) | 501 (56.9%) | 283 (61.7%) | <0.0001 |
| Calcium-blockers | 745 (34.3%) | 289 (31.5%) | 264 (30.0%) | 159 (34.6%) | <0.0001 |
| Oral anticoagulants | 493 (22.7%) | 360 (39.2%) | 350 (39.7%) | 215 (46.8%) | <0.0001 |
| RAS-inhibitors | 1105 (50.9%) | 500 (54.5%) | 497 (56.4%) | 246 (53.6%) | 0.02 |
| Statins | 1397 (64.3%) | 584 (63.6%) | 548 (62.2%) | 294 (64.1%) | 0.50 |
| **Year of procedure** | | | | | |
| 2008–2010 | 140 (6.4%) | 69 (7.5%) | 67 (7.6%) | 49 (10.7%) | 0.0001 |
| 2011–2013 | 330 (15.2%) | 156 (17.0%) | 163 (18.5%) | 73 (15.9%) | |
| 2014–2016 | 629 (29.0%) | 308 (33.6%) | 286 (32.5%) | 137 (29.9%) | |
| 2017–2019 | 1074 (49.4%) | 385 (41.9%) | 365 (41.4%) | 200 (43.6%) | |
| **Access** | | | | | |
| Transaortic | 65 (3.0%) | 37 (4.0%) | 34 (3.9%) | 15 (3.3%) | 0.07 |
| Transapical | 257 (11.8%) | 130 (14.2%) | 126 (14.3%) | 73 (15.9%) | |
| Transfemoral | 1851 (85.2%) | 751 (81.8%) | 721 (81.8%) | 371 (80.8%) | |

Abbreviations: TAVI, transcatheter aortic valve implantation; ADP, adenosin diphosphate; RAS, renin-angiotensin system.

Low loop diuretic treatment 1–40 mg of furosemide (or equivalent bumetanide); intermediate loop diuretic treatment 41–120 mg of furosemide; high loop diuretic treatment >120 mg furosemide.

## Factors associated with increased loop diuretic treatment

In order to outline what characterized patients, who had increased loop diuretic treatment after undergoing TAVI, we used a logistic regression analysis to identify available covariates associated with increased diuretic treatment compared with reduced/unchanged loop diuretic treatment. In the adjusted analysis, a history of ischemic heart disease, chronic heart failure,

**Table 2. Changes in loop diuretic treatment at 1-year post-TAVI.**

| Diuretic treatment at 1-year post-TAVI | Frequency (n) | Percent (%) |
|---|---|---|
| Increased loop diuretic treatment or hospitalization for heart failure | 1107 | 24.9 |
| Reduced loop diuretic treatment | 917 | 20.7 |
| Unchanged loop diuretic treatment | 2407 | 54.4 |

Abbreviations: TAVI, transcatheter aortic valve implantation.

Loop diuretic treatment pre-TAVI and 1-year post-TAVI (based on filled prescriptions) were assessed and grouped as receiving 1) low loop diuretic treatment 1–40 mg of furosemide (or equivalent bumetanide); intermediate loop diuretic treatment 41–120 mg of furosemide; high loop diuretic treatment >120 mg furosemide. Increased, reduced or unchanged loop diuretic treatment were defined as a change from one pre-TAVI loop diuretic group to a higher, lower, or unchanged loop diuretic group at 1-year post-TAVI. Hospitalization for heart failure was defined as having at least one readmission for heart failure within 1-year post-procedure.

atrial fibrillation, hypertension, and chronic kidney disease, prior to TAVI, were associated with increased loop diuretic treatment at 1-year post-TAVI (Fig 4).

## Subgroup analysis: Heart failure and loop diuretic treatment

In total, 1074 patients had a medical history with HF that were on diuretic treatment at time of TAVI procedure. The median age of the patients in this subgroup was 82.1 years ($25^{th}$-$75^{th}$ percentile 77–86) and 60% were men (Table 3). At time of TAVI procedure, 393 (36.6%) patients had low loop diuretic treatment, 398 (37.1%) had intermediate low loop diuretic treatment and 283 (26.4%) patients had high loop diuretic treatment. At 1-year post-TAVI, 248 (23.1%) patients had increased loop diuretic treatment, 478 (44.5%) had reduced loop diuretic treatment and 348 (32.4%) had unchanged loop diuretic treatment (S2 Fig in S1 File). The cumulative 5-year risk of death in patients undergoing TAVI, surviving one year, was 68.5% (95% CI: 59.7% to 75.8%) in patients with increased loop diuretic treatment and 61.6% (95% CI: 56.7% to 66.2%) in patients with reduced/unchanged loop diuretic treatment (Fig 5). In multivariable Cox proportional hazard analysis, increased loop diuretic treatment was associated with a higher risk of death (HR = 1.33, 95% CI 1.08–1.63) compared with reduced/unchanged loop diuretic treatment. In adjusted analysis, ischemic heart disease and atrial fibrillation were associated with increased loop diuretic treatment (S3 Fig in S1 File).

## Sensitivity analysis

For sensitivity purposes, we examined a combined outcome consisting of at least one hospital admission for HF within 1-year post-discharge for TAVI, increased loop diuretic treatment at 1-year post-TAVI or death after discharge date for TAVI procedure. Among all patients undergoing TAVI, 1107 (24.9%) patients died, had increased diuretic treatment or had at least one HF-admission. At 1-year post-TAVI, 917 (20.7%) patients had reduced loop diuretic treatment and 2407 (54.3%) patients had unchanged loop diuretic treatment.

## Discussion

This large nationwide observational cohort study examining loop diuretic treatment prior to TAVI and at 1-year post-TAVI demonstrated the following principal findings: (1) of all TAVI-patients, about half were treated with loop diuretics pre-TAVI and 1-year post-TAVI, (2) At 1-year post-TAVI, 893 (20%) had increased loop diuretic treatment, 1010 (23%) had reduced loop diuretic treatment and 2528 (57%) had unchanged loop diuretic treatment; (3) Our study

*A)*

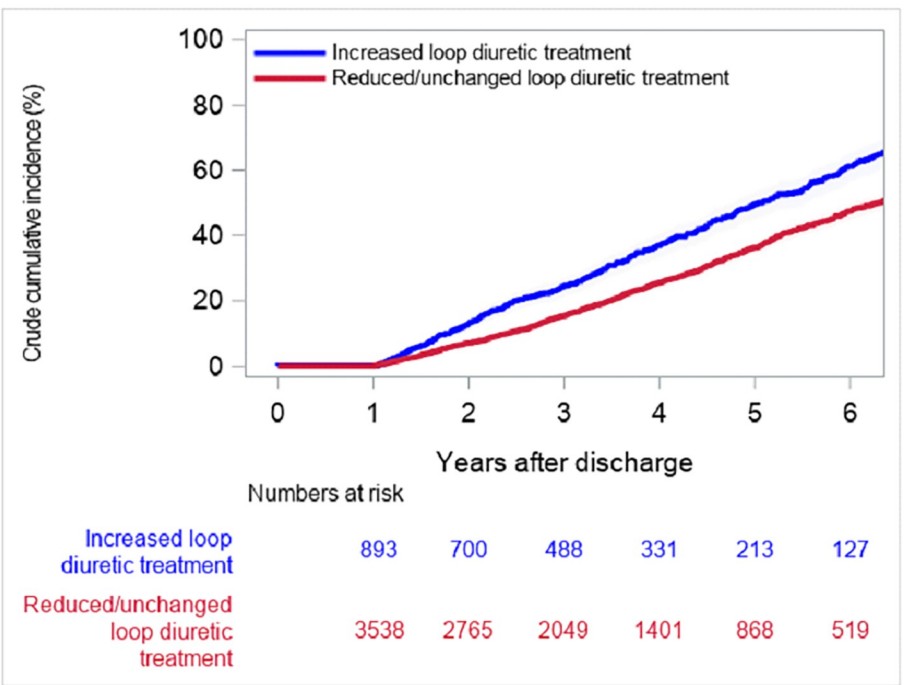

*B)*

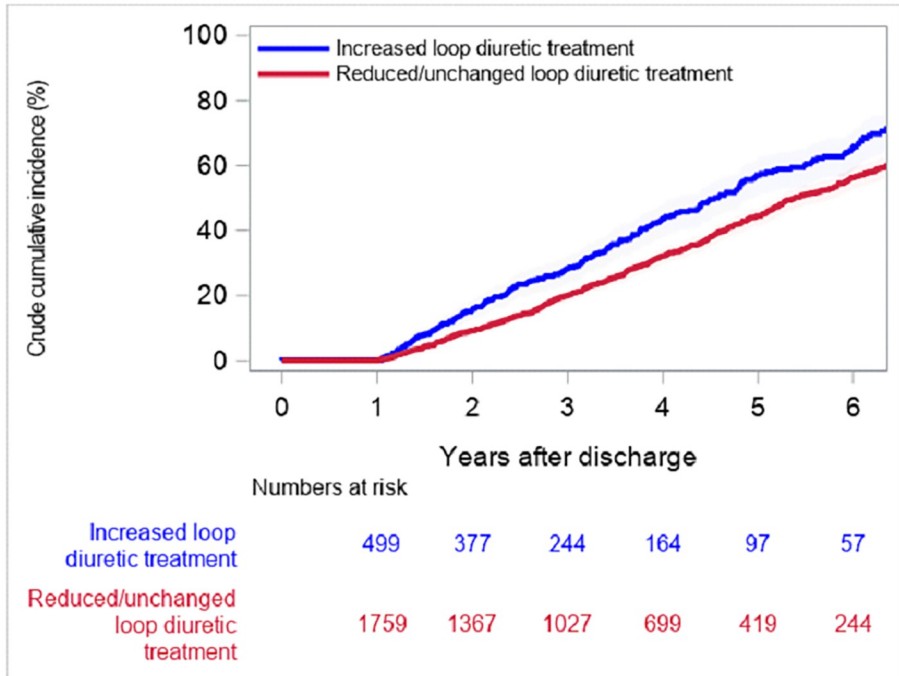

**Fig 3. Cumulative incidence of death among patients alive one year after discharge for TAVI procedure with increased and reduced/unchanged loop diuretic treatment at 1-year post-TAVI.** A) Patients alive one year after discharge for TAVI procedure. (B) Patients alive one year after discharge for TAVI procedure, excluding patients with no loop diuretic treatment pre-procedure. *Abbreviations*: *TAVI, transcatheter aortic valve implantation.*

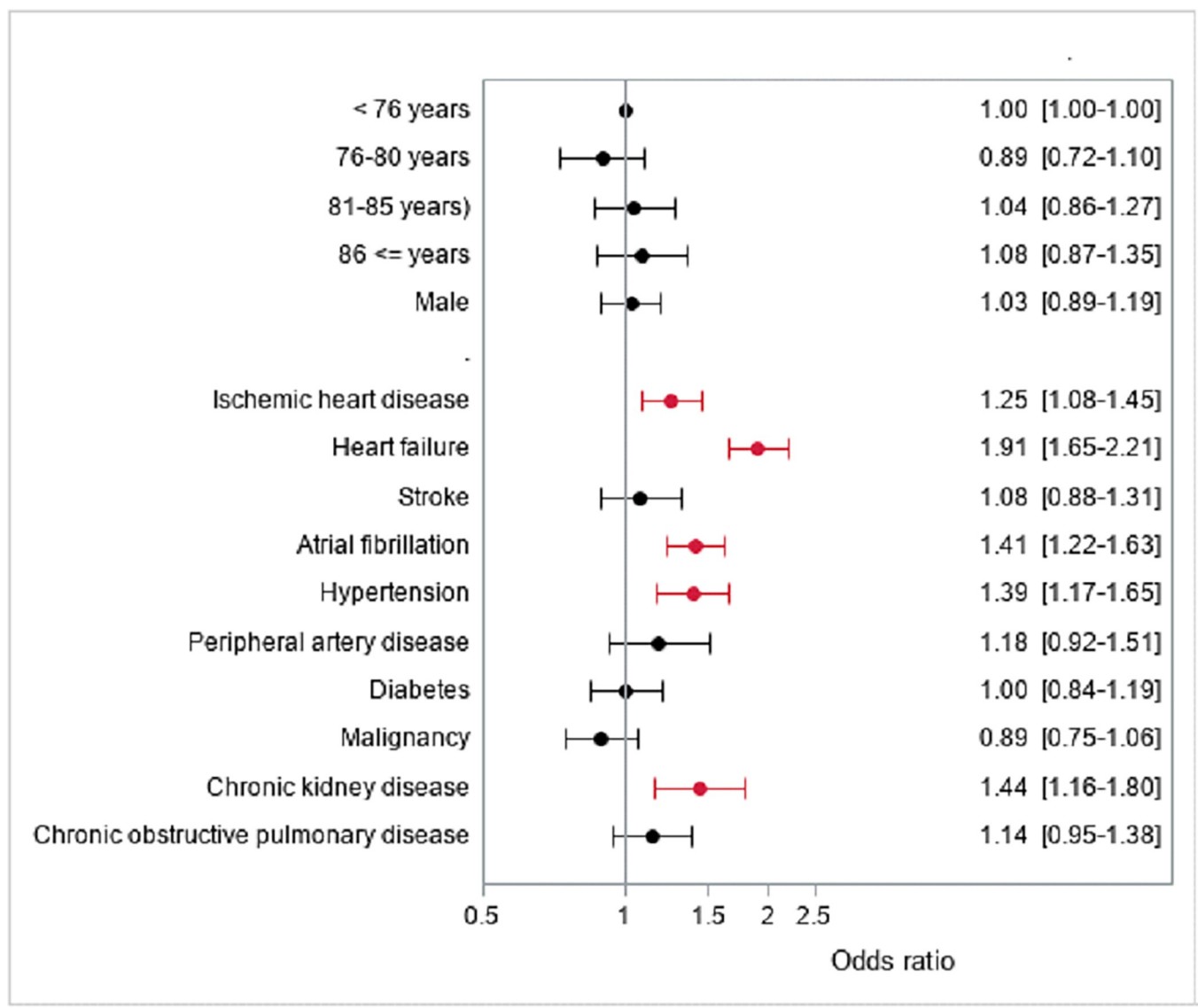

**Fig 4. Logistic regression of increased loop diuretic treatment at 1-year post-TAVI in patients undergoing TAVI.** Abbreviations: TAVI, transcatheter aortic valve implantation.

suggests that increased loop diuretic treatment is associated with a higher risk of death compared with reduced/unchanged loop diuretic treatment.

Prior studies have evaluated outcomes in AS-patients undergoing TAVI, with risk of death and rehospitalizations as main outcomes, suggesting TAVI to be a preferred treatment option, compared with medical treatment, in patients with symptomatic and severe AS [1, 2, 9– 11]. However, knowledge about quality of life and symptom burden is limited in these patients and actual patient-level symptoms-relief is not well studied. In our study, we compare loop diuretic treatment prior and at 1-year post-TAVI, as a proxy for symptom severity, in order to investigate if patients have symptom relief following intervention. Our findings illustrate that among patients with diuretic use pre-TAVI, 43% went from one pre-TAVI loop diuretic group to another post-TAVI loop diuretic group. However, the overall distribution of patients in each loop diuretic group pre-TAVI was similar to the distribution of patients in the loop diuretic

**Table 3. Baseline characteristics of patients with a medical history with heart failure and loop diuretic treatment at TAVI-procedure.**

| | Low loop diuretic treatment N = 393 | Intermediate loop diuretic treatment N = 398 | High loop diuretic treatment N = 283 | P-value |
|---|---|---|---|---|
| **Demographics** | | | | |
| Age, yrs | 82.2 (76.7–85.2) | 82.5 (77.2–86.2) | 81.2 (76.1–85.1) | 0.12 |
| Male | 253 (64.4%) | 221 (55.5%) | 172 (60.8%) | 0.24 |
| **Comorbidities** | | | | |
| Ischemic heart disease | 228 (58.0%) | 243 (61.1%) | 183 (64.7%) | 0.08 |
| Stroke | 57 (14.5%) | 60 (15.1%) | 45 (15.9%) | 0.62 |
| Atrial fibrillation | 196 (49.9%) | 203 (51.0%) | 168 (59.4%) | 0.02 |
| Peripheral vascular disease | 38 (9.7%) | 43 (10.8%) | 34 (12.0%) | 0.33 |
| Hypertension | 344 (87.5%) | 361 (90.7%) | 258 (91.2%) | 0.21 |
| Diabetes | 77 (19.6%) | 90 (22.6%) | 76 (26.9%) | 0.01 |
| Malignancy | 75 (19.1%) | 99 (24.9%) | 65 (23.0%) | 0.18 |
| Chronic kidney disease | 39 (9.9%) | 49 (12.3%) | 65 (23.0%) | <0.0001 |
| Chronic obstructive pulmonary disease | 73 (18.6%) | 87 (21.9%) | 61 (21.6%) | 0.31 |
| Liver disease | 6 (1.5%) | 13 (3.3%) | 9 (3.2%) | 0.15 |
| **Concomitant pharmacotherapy** | | | | |
| Thiazides | 99 (25.2%) | 67 (16.8%) | 49 (17.3%) | 0.01 |
| Spironolactone | 78 (19.9%) | 74 (18.6%) | 54 (19.1%) | 0.78 |
| Aspirin | 192 (48.9%) | 193 (48.5%) | 152 (53.7%) | 0.25 |
| ADP-receptor inhibitors | 92 (23.4%) | 98 (24.6%) | 72 (25.4%) | 0.54 |
| Beta-blockers | 223 (56.7%) | 252 (63.3%) | 187 (66.1%) | 0.01 |
| Calcium-blockers | 100 (25.4%) | 91 (22.9%) | 84 (29.7%) | 0.28 |
| Oral anticoagulants | 163 (41.5%) | 165 (41.5%) | 141 (49.8%) | 0.04 |
| RAS-inhibitors | 230 (58.5%) | 236 (59.3%) | 154 (54.4%) | 0.33 |
| Statins | 258 (65.6%) | 253 (63.6%) | 184 (65.0%) | 0.82 |
| **Year of procedure** | | | | |
| 2008–2010 | 29 (7.4%) | 41 (10.3%) | 40 (14.1%) | 0.15 |
| 2011–2013 | 70 (17.8%) | 68 (17.1%) | 39 (13.8%) | |
| 2014–2016 | 130 (33.1%) | 133 (33.4%) | 90 (31.8%) | |
| 2017–2019 | 164 (41.7%) | 156 (39.2%) | 114 (40.3%) | |
| **Access** | | | | |
| Transaortic | 15 (3.8%) | 11 (2.8%) | 5 (1.8%) | 0.30 |
| Transapical | 55 (14.0%) | 58 (14.6%) | 52 (18.4%) | |
| Transfemoral | 323 (82.2%) | 329 (82.7%) | 226 (79.9%) | |

Abbreviations: TAVI, transcatheter aortic valve implantation; ADP, adenosin diphosphate; RAS, renin-angiotensin system.

Low loop diuretic treatment 1–40 mg of furosemide (or equivalent bumetanide); intermediate loop diuretic treatment 41–120 mg of furosemide; high loop diuretic treatment >120 mg furosemide.

groups post-TAVI. Additionally, 418 patients (8%) died within 1-year post-procedure. In a recent study, Cantey et al. [14] examined loop diuretic treatment pre-TAVI according to outcomes following TAVI, which was the first study to investigate loop diuretic treatment and post-TAVI outcomes. Although few studies have evaluate diuretic treatment and quality of life, it seems reasonable for loop diuretic treatment to indicate the degree of dyspnea and furthermore is a proxy of the patient´s symptom burden, which can be used as an indicator of quality of life [22]. In our study, we found that in the subgroup of patients with HF and on diuretic treatment at baseline (TAVI procedure), were more likely to have either increased or

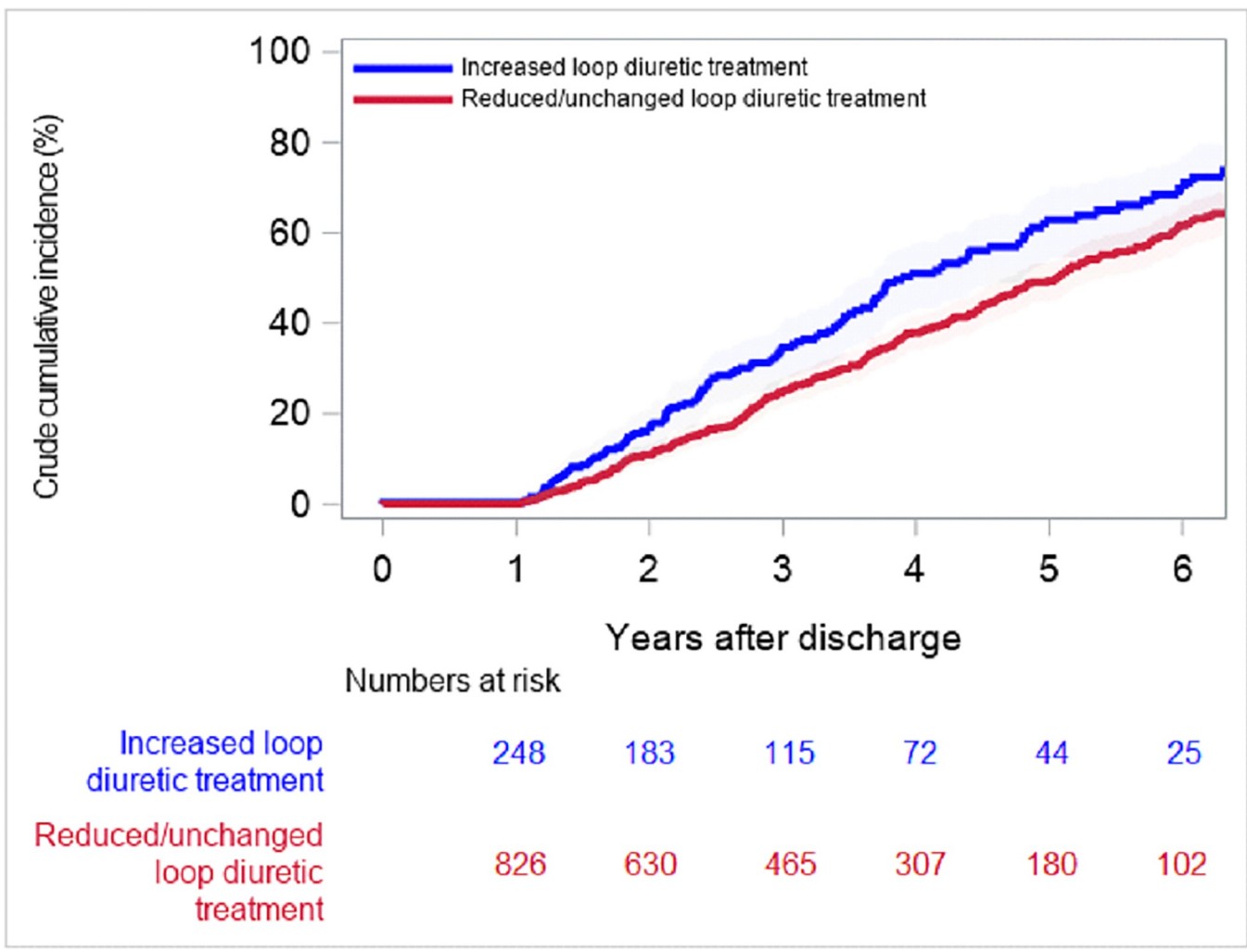

**Fig 5. Cumulative incidence of death among patients alive one year after discharge for TAVI procedure, with a medical history of heart failure and loop diuretic treatment post-procedure.** Abbreviations: TAVI, transcatheter aortic valve implantation.

reduced loop diuretic treatment following intervention compared to the overall study population. More specifically, 35% of patients with HF and loop diuretic treatment had increased loop diuretic treatment and 38% of patients had reduced loop diuretic treatment at 1-year post-TAVI. A possible explanation is that patients in the subgroup have more comorbidities and a greater symptom burden, meaning more vascular congestion and higher need for diuretic treatment. This statement can be supported by the fact, that the proportion of patients with high-dose loop diuretic treatment is higher in the subgroup with HF-patients and loop diuretic treatment, compared with the overall study population. Especially, there was a difference in the proportion of patients, who had intermediate-dose loop diuretic treatment pre-TAVI, which was 37% in the HF-subgroup compared to 17% in the overall study population. Furthermore, 26% of patients with HF and loop diuretic treatment had high-dose loop diuretic treatment, compared to 10% in the overall study population. In line with these results, the cumulative 5-year risk of death in patients surviving one year was higher in the HF-subgroup compared with the study population. However, the cumulative 5-year risk of death in patients surviving 1-year post-procedure was higher in patients with increased loop diuretic treatment

compared with reduced/unchanged loop diuretic treatment 1-year post-procedure in both groups. Additionally, increased loop diuretic treatment was associated with a greater risk of death compared with reduced/unchanged loop diuretic treatment 1-year post-procedure. This finding may reflect that patients, who use more loop diuretic treatment after undergoing TAVI, have a higher burden of comorbidities and symptoms, which causes more need for treatment and frailer patients. In addition, another study evaluated loop diuretic treatment and found an increased risk of all-cause mortality when using loop diuretic treatment, which also could be a possible explanation [23].

### Factors associated with increased loop diuretic treatment

This study suggests that ischemic heart disease, heart failure, atrial fibrillation, hypertension, and chronic kidney disease are associated with increased loop diuretic treatment 1-year post-TAVI. Ischemic heart disease, heart failure and atrial fibrillation may be explained by the fact, that the problem is myocardial ischemia, and therefore TAVI will not improve symptoms. According to chronic kidney disease, congestion and need for loop diuretic treatment is not due to cardiac but renal disease, which could explain the association. These prognostic factors associated could contribute to a better selection and understanding of patients, who have symptom relief after undergoing TAVI. However, this study illustrates associations in a broad perspective, and there is a need for more data to confirm these findings. Symptom relief may in fact be an important outcome—even patient reported—in future randomized studies.

### Strengths and limitations

This study was a nationwide cohort of patients undergoing TAVI, in a "real-life" setting with long-term follow-up. Additionally, in Denmark, pharmacies are required to register all redeemed prescriptions in order to complete reimbursement of drug expenses by the Danish heath care system, which makes the prescription registry very reliable. Limitations include factors inherent to all non-randomized, observational studies. Furthermore, data on important clinical parameters such as NYHA classification, description of symptom burden, echocardiographic findings and objective measurements of heart failure were not available. Neither did we have access to information on paravalvular regurgitation, mean gradient across the AV, nor the indication for starting loop diuretic treatment. Therefore, the calculation of treatment duration and dose is an approximation. Moreover, currently there is no specific guideline to evaluation of the loop diuretic treatment, which makes it uncertain how often, and if, doctors evaluate the loop diuretic treatment regularly. However, prescriptions of loop diuretics illustrate the presence of symptoms and need for symptom relief. Furthermore, our study suggest that loop diuretic treatment is a marker of higher-risk patients with more medical comorbidities.

## Conclusions

Among patients undergoing TAVI, surviving one year, one fifth of patients had increased loop diuretic treatment, and a little over one fifth of patients had reduced loop diuretic treatment 1-year post-procedure. In patients with increased diuretic treatment, the risk of death was higher compared to those with reduced/unchanged loop diuretic treatment.

## Supporting information

**S1 File.**
(DOCX)

## Author Contributions

**Supervision:** Jawad Haider Butt, Søren Lund Kristensen, Peter Ejvin Weeke, Ole De Backer, Morten Schou, Lars Køber, Emil Loldrup Fosbøl.

**Writing – original draft:** Xenia Begun.

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
