## [Decision Letter · Decision Letter 0]

29 Nov 2022

PONE-D-22-29562Diuretic treatment before and after Transcatheter Aortic Valve Implantation: A Danish nationwide studyPLOS ONE

Dear Dr. Begun,

Thank you for submitting your manuscript to PLOS ONE. After careful consideration, we feel that it does not fully meet PLOS ONE’s publication criteria as it currently stands. Therefore, we invite you to submit a revised version of the manuscript that addresses the points raised during the review process. The manuscript was reviewed by two experts in the field and they have provided some valuable comments and suggestions for you to consider and perhaps implement that will strengthen the manuscript. I agree with their assessment particularly about considering other hemodynamic and echocardiography parameters of heart failure and using them as part of multivariable model to adjust for them. I hope you will use their comments as constructive and submit a revised manuscript.

We look forward to receiving your revised manuscript.

Kind regards,

Jaimin R. Trivedi, MBBS, MPH

Academic Editor

PLOS ONE

5. Please include captions for your Supporting Information files at the end of your manuscript. Please see our Supporting Information guidelines for more information: http://journals.plos.org/plosone/s/supporting-information.

Reviewers' comments:

Reviewer's Responses to Questions

**Comments to the Author**

1. Is the manuscript technically sound, and do the data support the conclusions?

Reviewer #1: Yes

Reviewer #2: Yes

2. Has the statistical analysis been performed appropriately and rigorously? 

Reviewer #1: Yes

Reviewer #2: Yes

3. Have the authors made all data underlying the findings in their manuscript fully available?

Reviewer #1: Yes

Reviewer #2: Yes

4. Is the manuscript presented in an intelligible fashion and written in standard English?

Reviewer #1: Yes

Reviewer #2: Yes

5. Review Comments to the Author

Reviewer #1: Diuretic treatment before and after transcatheter aortic valve implantation: A Danish nationwide study

In this retrospective, registry based Danish study, the authors studied the value of loop diuretic use as a predictor for 1 year outcome in 4431 pts with severe aortic stenosis surviving 1 year after transcatheter aortic valve implantation (TAVI). Fifty % used diuretics at TAVI and 20% had increased dosage at 1 year. Diuretic use was associated with cardiovascular disease burden. By multivariate survival analysis they found that increased diuretic use at 1 year was significantly associated with poorer outcome. The paper is well written, methods are adequate utilising well designed health registries in Denmark, thus results seem robust. I have som general comments:

1. Study population was all pts surviving 1 year post TAVI, pts who died earlier were excluded (p.6). However, primary outcome was all-cause mortality at 1 year. This is somewhat confusing. Thus I wonder, who were eventually included in the survival analysis? Please explain.

2. Although inreased diuretic use was associated with poorer outcome, cardiovascular disease burden is the likely spurious background factor as indicated by Supplemental Table 1 and Fig 3. Was these important background parameters controlled for in the multivariate analysis? If so, what impact did e.g. ischemic heart disease, heart failure etc have on the results?

3. In my opinion, Suppl Table 1 and Suppl Fig 3 conveys important information that preferably could be included in the main article.

4. The conclusion in abstract and article should be modified slightly to ensure that the reader is not left with the impression that there is a causal relationship between diuretics use and outcome, at least this is not proven by this study.

Reviewer #2: Thank you for allowing me to review this very interesting article by Begun and colleagues, titled 'Diuretic treatment before and after Transcatheter Aortic Valve Implantation: A Danish nationwide study'. The authors have used escalation of diuretic therapy as a surrogate marker for symptomatic non-relief, and have gone on to conclude that patients with increased diuretic use in the first year post TAVR, have reduced survival.

I find this article very interesting, and agree with the authors when they say that TAVR may/may not provide symptomatic relief to patients, and it is imperative to analyze this question further, specially with the surgical landscape changing towards TAVR in younger individuals. I have a few questions/comments

1. It would be interesting to know how the echo looked in these patients when they came in for follow-up. The reason i say that is, a significant number of these patients have varying degrees of mitral regurgitation. While the TAVR procedure takes care of the AS, the theory is that the MR should get better over time. But, often that is not the case. And patients with significant MR will have symptoms of dyspnea in the post-op period and may need escalating diureses. So, is the increase in diuresis due to symptomatic non relief from the AS, or progression of the MR, or simply progression of the heart failure? Echo findings would better corroborate your answer.

2. How many patients left the TAVR room with mild or greater paravalvar regurgitation? How many patients left the TAVR room with a double digit mean gradient across the AV. This may also have a role to play in heart failure admissions post-op.

3. 61% of the patients with high diuretic treatment had heart failure. What was the authors definition of heart failure?

4. Were their any objective measurements of heart failure? CXR, BNP? Curious because an 80 year old coming in with pneumonia if often also given an extra dose of diuretic since it is assumed that some of the pulmonary symptoms are attributed to diastolic dysfunction.

6. PLOS authors have the option to publish the peer review history of their article (what does this mean?). If published, this will include your full peer review and any attached files.

Reviewer #1: No

Reviewer #2: No

---

## [Author Response · Author response to Decision Letter 0]

6 Jan 2023

Reviewer #1:

In this retrospective, registry based Danish study, the authors studied the value of loop diuretic use as a predictor for 1 year outcome in 4431 pts with severe aortic stenosis surviving 1 year after transcatheter aortic valve implantation (TAVI). Fifty % used diuretics at TAVI and 20% had increased dosage at 1 year. Diuretic use was associated with cardiovascular disease burden. By multivariate survival analysis they found that increased diuretic use at 1 year was significantly associated with poorer outcome. The paper is well written, methods are adequate utilizing well designed health registries in Denmark, thus results seem robust. I have some general comments:

Comment no. 1: Study population was all pts surviving 1 year post TAVI, pts who died earlier were excluded (p.6). However, primary outcome was all-cause mortality at 1 year. This is somewhat confusing. Thus I wonder, who were eventually included in the survival analysis? Please explain.

Our response: Thank you for your comment. It is correct that all patients, who died within the first year post TAVI-discharge were excluded in the study. Hence, index time for the analysis of mortality was set at one year post-TAVI. The primary outcome was all-cause mortality among patients, who survived one year post-procedure and then to one year post this index time, and this has also been specified in the manuscript. We have replaced the figures to the ones shown below. If the editors like the latter version better, we will change this.

Changes made to the manuscript:

Methods section, p. 7, lines 129-130:

Outcomes 

The primary outcomes were change in loop diuretic treatment pre-TAVI and at 1-year post-TAVI and all-cause mortality among patients alive one year after discharge for TAVI procedure.

Figure 3: Cumulative incidence of death among patients alive one year after discharge for TAVI procedure with increased and reduced/unchanged loop diuretic treatment intensity at 1-year post-TAVI

 A) Patients alive one year after discharge for TAVI procedure 

(B) Patients alive one year after discharge for TAVI procedure, excluding patients with no loop diuretic treatment pre-procedure

Comment no. 2: Although increased diuretic use was associated with poorer outcome, cardiovascular disease burden is the likely spurious background factor as indicated by Supplemental Table 1 and Fig 3. Was these important background parameters controlled for in the multivariate analysis? If so, what impact did e.g. ischemic heart disease, heart failure etc have on the results?

Our response: Thank you for your comment. In our multivariable Cox proportional hazard regression model, we adjusted for a history of heart failure and ischemic heart disease. We also adjusted for age (included as a categorical variable: <76 years, 76-80 years, 81-85 years and 86 <= years), year of procedure (included as a categorical variable: 2008-2010, 2011-2013, 2014-2016 and 2017-2019), sex, a history of stroke, atrial fibrillation, hypertension, peripheral vascular disease, diabetes, malignancy, chronic kidney disease, chronic obstructive pulmonary disease, and liver disease. In our adjusted multivariable Cox proportional hazard analysis, we found that increased loop diuretic treatment was associated with a higher risk of death (HR=1.36, 95% CI 1.22-1.52) compared with reduced/unchanged loop diuretic treatment. Additionally, we did a multivariable Cox proportional hazard analysis for patients, who were on diuretic treatment pre-TAVI, and found that increased loop diuretic treatment was associated with a higher risk of death (HR=1.31, 95% CI 1.13-1.52) compared with reduced/unchanged loop diuretic treatment. Hence, adjustment for these factors reduced but not removed the higher risk associated with increased loop diuretic treatment.

In our subgroup analysis, consisting of patients, who had a medical history with HF that were on diuretic treatment at time of TAVI procedure, we also did a multivariable Cox proportional hazard analysis, and found that increased loop diuretic treatment was associated with a higher risk of death (HR=1.33, 95% CI 1.08-1.63) compared with reduced/unchanged loop diuretic treatment. Hence, we tried to adjust our estimates as best as possible and we still found a significant relationship between loop diuretic use and mortality. 

Comment no. 3: In my opinion, Suppl Table 1 and Suppl Fig 3 conveys important information that preferably could be included in the main article.

Our response: Thank you for your suggestion. We have now included Supplementary Table 1 and Supplementary Figure 3 in the main article, as Table 3 and Figure 5, respectively. 

Comment no. 4: The conclusion in abstract and article should be modified slightly to ensure that the reader is not left with the impression that there is a causal relationship between diuretics use and outcome, at least this is not proven by this study.

Our response: We appreciate your comment. We have modified the conclusion, so it is not the increased diuretic treatment, which was associated with a higher risk of death, but the associated risk of death was higher in patients with increased diuretic treatment.

Changes made to the manuscript:

Conclusions: Among patients undergoing TAVI, surviving one year, one fifth of patients had increased loop diuretic treatment, and a little over one fifth had reduced loop diuretic treatment 1-year post-procedure. In patients with increased diuretic treatment, the associated risk of death was higher compared with reduced/unchanged loop diuretic treatment. 

Reviewer #2:

Thank you for allowing me to review this very interesting article by Begun and colleagues, titled 'Diuretic treatment before and after Transcatheter Aortic Valve Implantation: A Danish nationwide study'. The authors have used escalation of diuretic therapy as a surrogate marker for symptomatic non-relief, and have gone on to conclude that patients with increased diuretic use in the first year post TAVR, have reduced survival.

I find this article very interesting, and agree with the authors when they say that TAVR may/may not provide symptomatic relief to patients, and it is imperative to analyze this question further, specially with the surgical landscape changing towards TAVR in younger individuals. I have a few questions/comments.

Comment no. 1: It would be interesting to know how the echo looked in these patients when they came in for follow-up. The reason i say that is, a significant number of these patients have varying degrees of mitral regurgitation. While the TAVR procedure takes care of the AS, the theory is that the MR should get better over time. But, often that is not the case. And patients with significant MR will have symptoms of dyspnea in the post-op period and may need escalating diureses. So, is the increase in diuresis due to symptomatic non relief from the AS, or progression of the MR, or simply progression of the heart failure? Echo findings would better corroborate your answer.

Our response: We value this comment, but unfortunately, we do not have access to the echo findings nor to the indication for prescription of diuretic medication. The lack of data on echo findings has now been acknowledged in the Limitations section. 

Changes made to the manuscript:

Limitations section, p. 15, lines 287-289: 

Furthermore, data on important clinical parameters such as NYHA classification, description of symptom burden, echocardiographic findings and the indication for starting loop diuretic treatment, were not available.

Comment no. 2: How many patients left the TAVR room with mild or greater paravalvar regurgitation? How many patients left the TAVR room with a double digit mean gradient across the AV. This may also have a role to play in heart failure admissions post-op.

Our response: Again, the reviewer raises an important issue, but unfortunately, we do not have access to the clinical data in this retrospective, registry-based study. The lack of data on paravalvular regurgitation and mean gradient have now been acknowledged in the Limitations section. 

Changes made to the manuscript:

Limitations section, p. 15, lines 287-289: 

Furthermore, data on important clinical parameters such as NYHA classification, description of symptom burden, echocardiographic findings, information on paravalvular regurgitation or mean gradient across the AV, and the indication for starting loop diuretic treatment, were not available.

Comment no. 3: 61% of the patients with high diuretic treatment had heart failure. What was the authors definition of heart failure?

Our response: Thank you for your comment. Comorbidities were obtained by identifying diagnoses during hospitalizations any time before discharge date after the TAVI procedure and admissions were identified by using ICD-8 and ICD-10 diagnosis codes. Heart failure was defined by having a discharge diagnosis consisting of one or more of the codes ICD-in regarding heart failure, cardiomyopathy, hypertensive heart disease or pulmonary edema (diagnosis code DI50, DI42, DJ819, DI110, DI130, DI132, DI425, DI428). This has been validated by another study, and have a very high positive predictive value in the registry (Olesen et al., 2011).

Comment no. 4: Were their any objective measurements of heart failure? CXR, BNP? Curious because an 80 year old coming in with pneumonia if often also given an extra dose of diuretic since it is assumed that some of the pulmonary symptoms are attributed to diastolic dysfunction.

Our response: We value this comment, which is important. Unfortunately, we do not have access to objective measurements of heart failure and the indication for diuretic treatment in this study. We agree that this could be the case of increased diuretic treatment, however, to eliminate uncertainties, we have divided loop diuretic treatment into groups according to daily dose of the claimed drug: 1) low dose: 1-40 mg of furosemide or <=1 mg of bumetanide; 2) intermediate: 41-120 mg of furosemide or 2-3 mg bumetanide and 3) high: >120 mg furosemide or >3 mg bumetanide. Additionally, the daily dose was calculated according to the information on the prescriptions from The National Prescription Registry. Five prescriptions prior to the actual prescription were assessed and based on dispensed number of pills and strength of the prescription, a treatment interval was constituted, which was used to estimate the daily dose. However, we have acknowledged the lack of data on objective measurements of heart failure in the Limitations section. 

Changes made to the manuscript:

Limitations section, p. 15, lines 287-289: 

Furthermore, data on important clinical parameters such as NYHA classification, description of symptom burden, echocardiographic findings and objective measurements of heart failure were not available. Neither did we have access to information on paravalvular regurgitation, mean gradient across the AV, nor the indication for starting loop diuretic treatment.

 

References

Olesen, J. B., Lip, G. Y. H., Hansen, M. L., Hansen, P. R., Tolstrup, J. S., Lindhardsen, J., Selmer, C., Ahlehoff, O., Olsen, A. M. S., Gislason, G. H., & Torp-Pedersen, C. (2011). Validation of risk stratification schemes for predicting stroke and thromboembolism in patients with atrial fibrillation: nationwide cohort study. BMJ (Clinical Research Ed.), 342(7792), 320. https://doi.org/10.1136/BMJ.D124

---

## [Decision Letter · Decision Letter 1]

23 Jan 2023

PONE-D-22-29562R1Diuretic treatment before and after Transcatheter Aortic Valve Implantation: A Danish nationwide studyPLOS ONE

Dear Dr. Begun,

Thank you for submitting your revised manuscript to PLOS ONE. After careful consideration, we feel that it has merit but a few minor points still need to be addressed to meet PLOS ONE’s publication criteria as it currently stands. Therefore, we invite you to submit a revised version of the manuscript that addresses the points raised during the review process.

All the reviewers have re-evaluated your revised manuscript and have provided positive feedback however, one reviewer has some additional comments for you to address. 

We look forward to receiving your revised manuscript.

Kind regards,

Jaimin R. Trivedi, MBBS, MPH

Academic Editor

PLOS ONE

Journal Requirements:

Additional Editor Comments:

The authors have addressed majority of concerns; however, few minor points still remain before acceptance.

Reviewers' comments:

Reviewer's Responses to Questions

**Comments to the Author**

1. If the authors have adequately addressed your comments raised in a previous round of review and you feel that this manuscript is now acceptable for publication, you may indicate that here to bypass the “Comments to the Author” section, enter your conflict of interest statement in the “Confidential to Editor” section, and submit your "Accept" recommendation.

Reviewer #1: (No Response)

Reviewer #2: All comments have been addressed

2. Is the manuscript technically sound, and do the data support the conclusions?

Reviewer #1: Yes

Reviewer #2: Partly

3. Has the statistical analysis been performed appropriately and rigorously? 

Reviewer #1: Yes

Reviewer #2: Yes

4. Have the authors made all data underlying the findings in their manuscript fully available?

Reviewer #1: Yes

Reviewer #2: Yes

5. Is the manuscript presented in an intelligible fashion and written in standard English?

Reviewer #1: Yes

Reviewer #2: Yes

6. Review Comments to the Author

Reviewer #1: Diuretic treatment before and after Transcatheter Aortic Valve Implantation: A Danish nationwide study

Resubmission #1

I thank the authors for their swift and adequate reply to my comments which improved my understanding of this issue. I merely have a couple of suggestions:

Comment:

1. I still think the conclusion in Abstract and main text could be improved somewhat, e.g. instead of "In patients with increased diuretic treatment, the associated risk of death was higher compared with reduced/unchanged loop diuretic treatment" it may be better to simplify to "In patients with increased diuretic treatment, the risk of death was higher compared to those with reduced/unchanged loop diuretic treatment."

2. If pressed for space, Figure 3 B could probably be omitted, and results rather be included in the ms text.

Minor:

PP12: atrial fibrillation not arterial fibrillation

Reviewer #2: The authors need to be congratulated on attempting a nation-wide study to look at diuretic need after TAVR as a marker for worse outcome. The major limitations of this manuscript are that echocardiographic data on post deployment para-valvar leak, or degree of mitral regurgitation was not available, as these can be confounding factors while assessing diuretic need post TAVR.

Nonetheless, I feel this publication has value, and would add to the existing literature.

7. PLOS authors have the option to publish the peer review history of their article (what does this mean?). If published, this will include your full peer review and any attached files.

Reviewer #1: No

Reviewer #2: No

---

## [Author Response · Author response to Decision Letter 1]

15 Feb 2023

Reviewer #1:

I thank the authors for their swift and adequate reply to my comments which improved my understanding of this issue. I merely have a couple of suggestions:

Comment no. 1: I still think the conclusion in Abstract and main text could be improved somewhat, e.g. instead of "In patients with increased diuretic treatment, the associated risk of death was higher compared with reduced/unchanged loop diuretic treatment" it may be better to simplify to "In patients with increased diuretic treatment, the risk of death was higher compared to those with reduced/unchanged loop diuretic treatment."

Our response: We would like to thank the reviewer for this comment. We have modified the conclusion to avoid any impression that there is a relationship between diuretics and outcome. 

Changes made to the manuscript:

Conclusions: Among patients undergoing TAVI, surviving one year, one fifth of patients had increased loop diuretic treatment, and a little over one fifth had reduced loop diuretic treatment 1-year post-procedure. In patients with increased diuretic treatment, the associated risk of death was higher compared with the risk of death was higher compared to those with reduced/unchanged loop diuretic treatment. 

Comment no. 2: If pressed for space, Figure 3 B could probably be omitted, and results rather be included in the ms text.

Our response: We thank the reviewer for this comment. We agree with the reviewer, that this figure is not central in the manuscript. However, we suggest that if editor is pressed for space, we will move the figure to supplemental data. 

Minor: PP12: atrial fibrillation not arterial fibrillation

Our response: Thank you for this comment. We have corrected the manuscript as shown below. 

Changes made to the manuscript:

Results section, p. 12, lines 215:

In adjusted analysis, ischemic heart disease and arterial atrial fibrillation were associated with increased loop diuretic treatment (Supplementary figure 3). 

Reviewer #2:

The authors need to be congratulated on attempting a nation-wide study to look at diuretic need after TAVR as a marker for worse outcome. The major limitations of this manuscript are that echocardiographic data on post deployment para-valvar leak, or degree of mitral regurgitation was not available, as these can be confounding factors while assessing diuretic need post TAVR.

Nonetheless, I feel this publication has value, and would add to the existing literature.

Our response: We appreciate this comment made by the reviewer.

---

## [Editor Report · Decision Letter 2]

20 Feb 2023

Diuretic treatment before and after Transcatheter Aortic Valve Implantation: A Danish nationwide study

PONE-D-22-29562R2

Dear Dr. Begun,

We’re pleased to inform you that your manuscript has been judged scientifically suitable for publication and will be formally accepted for publication once it meets all outstanding technical requirements.

Kind regards,

Jaimin R. Trivedi, MBBS, MPH

Academic Editor

PLOS ONE
---

## [Editor Report · Acceptance letter]

6 Mar 2023

PONE-D-22-29562R2 

Diuretic treatment before and after Transcatheter Aortic Valve Implantation: A Danish nationwide study 

Dear Dr. Begun:

I'm pleased to inform you that your manuscript has been deemed suitable for publication in PLOS ONE. Congratulations! Your manuscript is now with our production department. 

Kind regards, 

on behalf of

Dr. Jaimin R. Trivedi 

Academic Editor

PLOS ONE